# Current Insights into Oral Cancer Diagnostics

**DOI:** 10.3390/diagnostics11071287

**Published:** 2021-07-16

**Authors:** Yee-Fun Su, Yi-Ju Chen, Fa-Tzu Tsai, Wan-Chun Li, Ming-Lun Hsu, Ding-Han Wang, Cheng-Chieh Yang

**Affiliations:** 1iStat Biomedical Co., Ltd., New Taipei City 221, Taiwan; yeefun@istat.com.tw (Y.-F.S.); coco@istat.com.tw (Y.-J.C.); 2Institute of Oral Biology, School of Dentistry, National Yang Ming Chiao Tung University, Taipei 11221, Taiwan; karentsai@ym.edu.tw (F.-T.T.); wcli@ym.edu.tw (W.-C.L.); 3Department of Dentistry, School of Dentistry, National Yang Ming Chiao Tung University, Taipei 11221, Taiwan; mlhsu@ym.edu.tw (M.-L.H.); nn2399906@ym.edu.tw (D.-H.W.); 4Oral and Maxillofacial Surgery, Department of Stomatology, Taipei Veterans General Hospital, Taipei 11217, Taiwan

**Keywords:** oral cancer, OPMD, diagnostic, early detection, biomarker, in vitro diagnostic, medical device

## Abstract

Oral cancer is one of the most common head and neck malignancies and has an overall 5-year survival rate that remains below 50%. Oral cancer is generally preceded by oral potentially malignant disorders (OPMDs) but determining the risk of OPMD progressing to cancer remains a difficult task. Several diagnostic technologies have been developed to facilitate the detection of OPMD and oral cancer, and some of these have been translated into regulatory-approved in vitro diagnostic systems or medical devices. Furthermore, the rapid development of novel biomarkers, electronic systems, and artificial intelligence may help to develop a new era where OPMD and oral cancer are detected at an early stage. To date, a visual oral examination remains the routine first-line method of identifying oral lesions; however, this method has certain limitations and as a result, patients are either diagnosed when their cancer reaches a severe stage or a high-risk patient with OPMD is misdiagnosed and left untreated. The purpose of this article is to review the currently available diagnostic methods for oral cancer as well as possible future applications of novel promising technologies to oral cancer diagnosis. This will potentially increase diagnostic options and improve our ability to effectively diagnose and treat oral cancerous-related lesions.

## 1. Introduction

Oral cancer is one of the most common head and neck cancers and is the eleventh most common cancer globally, with approximately 350,000 new cases and 177,000 deaths every year [1,2]. Two-thirds of global incidence occurs in low-income and middle-income countries (LMICs) with half of those cases occurring in South Asia [3,4]. The 5-year survival rate of oral cancer ranges from 15% to 50%, mainly because most oral cancers have progressed to an advanced stage, namely stage III or IV, at the time of diagnosis [5]. By way of contrast, the survival rate of oral cancer can exceed 80% if it is diagnosed and treated early, namely at stage I or stage II [6,7]. Oral squamous cell carcinoma (OSCC) arises from the epithelial layer, and accounts for more than 90% of oral malignant disease. The main sites of oral cancer are the lip, tongue, gum, floor of the mouth, palate, and a range of other parts of the oral cavity; these are generally categorized according to the International Classification of Disease (ICD) [8].

The risk factors for oral cancer vary depending on the geographical region and lifestyle habits of the region’s inhabitants. In Western countries, smoking and drinking are the major risk factors. This differs from South Asia and Pacific countries, where betel-nut chewing and smoking are the major factors [9]. Other factors such as HPV infection are also important causes of oral cancer [10,11,12,13,14].

Oral cancer is preceded by precancerous lesion that is called an oral potentially malignant disorder (OPMD), a term recommended by WHO classification [15]. OPMDs are a heterogeneous group of oral mucosal lesions associated with an increased risk of malignant transformation to cancer. Leukoplakia, erythroleukoplakia, oral submucous fibrosis, proliferative verrucous leukoplakia (PVL), and erythroplakia are among the most encountered OPMDs in clinical practice [16].

Early detection and evaluation of the potential risk of malignant transformation of an OPMD are crucial for physicians to monitor and treat oral cancer and improve the survival rate. At present, a visual oral examination (VOE) is the routine screening method used to identify oral mucosal lesions. However, VOE depends heavily on the experience of the physician, owing to the fact that several OPMDs, such as white or red lesions and persistent ulcers, are often indistinguishable at its clinical presentation [17]. A summary of the global oral cancer forum in 2016 depicted the shortcomings of VOE, in which the result of VOE is dependent on the quality of the examiner, requires training and calibration of screeners, and is indistinguishable from benign lesions, cancer, OPMDs, etc. [18].

Currently, biopsy of the representative tissues followed by histopathological assessment remains the gold standard for differentiating and diagnosing different types of OPMDs and oral cancers. Among all OPMDs, oral epithelial dysplasia is a more advanced lesion type that is characterized by its histological features. These include poor epithelial layer differentiation and maturation derangement, which result in a growth anomaly that is due to abnormal or atypical epithelial proliferation [19]. The WHO classifies dysplasia into mild, moderate, and severe, with the last including carcinoma in situ. This is the mainstream classification used by pathologists for histological diagnoses, despite the presence of several proposed classification systems [20]. The presence of oral epithelial dysplasia always carries a higher risk of malignant transformation, and OPMDs with oral epithelial dysplasia transform into carcinomas more frequently than OPMDs without dysplasia [21,22,23]. Despite the well-established and systemic approaches of histopathologic examination, some clinical conditions may face the clinician with great challenges in determining an accurate diagnosis. First, in a carcinogen-exposed oral cavity, the suspicious lesions commonly span across a large area or scatter at multiple loci. The mucosa affected by the effects of field cancerization or several lesions under various cancer-developing status may require multiple biopsies. Nevertheless, due to its invasiveness, many patients fear repeated biopsies, regardless of the presence of symptomatic or asymptomatic lesions. In addition, for some well-differentiated cancers or verrucous leukoplakias, the thickness and heavy keratinization of the lesions usually hinder the biopsy procedures to obtain a deep enough representative tissue specimen. Therefore, histopathological diagnosis heavily relies on the location, size, depth, and quality of the biopsied specimen, the techniques of fixation and freezing used on the biopsy, and the physician’s professional background and experience. The clinical observer’s subjectivity and variability during the VOE and histopathological examination poses limitations on OPMD detection and oral cancer diagnosis [24,25].

Early detection of OPMDs and oral cancer is of the utmost importance, and this facilitates diagnosis, treatment, and monitoring of the disease. It is especially critical to differentiate the lesions with higher malignant transformation rates because this will greatly influence the treatment outcome. Though histopathological appraisal of OPMDs remains the gold standard for clinical diagnosis and treatment, to differentiate lesions requiring active treatment may be insufficient, especially for those that are obscure or in a non-dysplasia state. The introduction of promising intracellular biomarkers may provide a better opportunity to precisely evaluate and predict the possibility and risk of OPMDs developing into an oral cancer lesion, which mandates active treatment when detected. The aim of this article is to provide a concise view of oral cancer diagnostic methods other than the traditional VOE or classical histopathological evaluation that are currently available in the clinical practice or new methods under optimization for clinical application. Future technologies that remain in the development stage are also introduced (Figure 1).

## 2. Current Available Techniques

### 2.1. Common Diagnostic Test Currently Available in Clinical Practice

There are several diagnostic methods that are used routinely in clinical practice, these include vital staining, oral cytology, and optical imaging technology.

#### 2.1.1. Vital Staining

Vital staining is a conventional tissue staining method that uses toluidine blue, a metachromatic dye, to stain cells with an increased DNA content and dysplastic or malignant cells containing abnormal DNA [26].

Toluidine blue has been shown to have high sensitivity, but relatively low specificity [27,28]. A recent hospital-based diagnostic accuracy study was carried out to evaluate the efficacy of Toluidine blue staining, which was served as an adjunct tool to standard clinical examination to facilitate early detection of oral cavity and oropharynx malignant lesions. Fifty-five subjects with OPMDs or malignant lesions were subjected to detailed clinical examination and toluidine blue staining. By comparing the staining results with histopathologic examination, the toluidine blue test detected malignancy with a sensitivity of 92.6% and a specificity of 67.9%. The overall diagnostic accuracy was 80%. The result indicates that the toluidine blue staining method can be a valuable adjunctive diagnostic process for oral and oropharyngeal cancers [27]. In practice, it is a useful way of identifying lesions with possible malignant changes.

Lugol’s iodine staining is another oral mucosa stain that can be used to delineate the oral mucosa by presenting a brownish or mahogany color when reacting with the glycogen present in normal oral mucosa. In contrast, Lugol’s iodine results in no staining and a pale appearance when a dysplastic or cancerous lesion is present, compared to the surrounding normal tissue [29,30]. A combination of Lugol’s iodine staining and toluidine blue staining can help distinguish the normal from abnormal epithelium and is particularly useful when selecting target sites for biopsy prior to treatment when there is a wide field of lesions [30,31].

#### 2.1.2. Oral Cytology

Exfoliative oral cytology is a conventional method that collects oral mucosal cells by scrapping, brushing, or rinsing the exfoliative cells using a tongue blade or brush [32]. The collected oral mucosal cell specimens are then fixed and stained, and their morphology is examined and interpreted by an experienced pathologist under a microscope [33]. This method is derived from a cervical pap smear, and is simple, non-aggressive, and relatively painless [34,35]. Oral cytology was first utilized to evaluate human oral mucosal lesions in 1963. However, as a screening method for oral precancer and oral cancer, it has not achieved the same success as that of cervical cancer screening [36]. It has been shown to exhibit low sensitivity in the diagnosis of oral cancer [37,38]. This may attribute to inadequate or nonrepresentative sampling, a high risk of procedural errors, or subjectivity of interpretation by examiners [39].

Over the years, oral cytology has undergone substantial improvements for early assessment of suspicious oral lesions. Brush cytology is also a method that can be applied to individuals who have difficulties opening their mouth for scalpel biopsies, which confirm a lesion site [40]. The OralCDx^®^ Brush Test (CDx Diagnostics, New York, NY, USA) is a minimally invasive brush biopsy that is coupled with an artificial intelligence computer-assisted tissue analysis. The term “brush biopsy” is argued to be replaced by the term “oral brush cytology” as this technique should be used as a complement test and not a replacement for biopsy [41]. OralCDx^®^ is an advanced complementary form of exfoliative oral cytology and has an adjunctive role as a diagnostic or screening tool for the identification of OPMD at an early stage [42]. However, the clinical performance of the OralCDx^®^ Brush Test remains controversial as the test’s results has been found to vary significantly across several studies. Several studies reported that OralCDx^®^ has high sensitivity and specificity in the detection of OSCC and precancerous lesions compared to a surgical biopsy [42,43,44]. However, there are also studies that questioned the previously reported high performance of the OralCDx^®^ system [44,45,46].

#### 2.1.3. Optical Imaging

Light-based technologies for detecting OPMDs and oral cancer have been used in clinical practice for several years. Chemiluminescence or fluorescent light are used as an intraoral detector to pinpoint oral mucosal lesions. The color of the light reflected from the oral epithelium is used to assess the condition of the oral epithelium [47].

##### Autofluorescence-Based

Autofluorescence refers to when light of a particular wavelength interacts with cells and causes excitation and re-emission of light of different wavelengths. Autofluorescence emitted from tissues is produced by fluorophores that naturally occurs in the human tissues, for instance collagen, tryptophan, elastin, keratin, hemoglobin, and NADH, and which are the naturally occurring fluorophores. The concentration of these fluorophores alters in OPMDs and cancerous lesions, which cause alterations of natural light scattering and absorption properties of the tissues [48].

VELScope^®^Vx (LED Dental, British Columbia, BC, Canada) is a CE-approved medical device that noninvasively screens for alterations in oral mucosal autofluorescence [49]. It is a handheld camera device that emits blue light (400 nm and 460 nm wavelengths) that is combined with proprietary optical filtering in order to visualize oral abnormalities. The normal oral mucosa produces green autofluorescent light, while anomalous oral mucosal lesions absorb the autofluorescent light and appear as dark areas that contrast with the adjacent tissue [49,50]. One clinical study has shown that use of the VELScope^®^Vx is beneficial for confirming the presence of OPMDs, including erythroplakia, leukoplakia, and other oral tissue disorders; however, it was unable to discriminate between low-risk and high-risk lesions [51]. Another study found that the VELScope^®^Vx alone performed relatively poorly when detecting epithelial dysplasia; albeit, while elevated oral mucosal lesions were visible and during the process, a few clinically undetected lesions were uncovered. This suggests the VELScope^®^Vx is more suitably used when accompanied by relevant clinical interpretation or by other methods for the diagnosis of epithelial dysplasia [52].

##### Chemiluminesence-Based

Chemiluminescence refers to the emission of the visible range of light radiation after the electrons, excited by a chemical exergonic reaction, return from an excited to ground state; light photons are released upon the transition of electronic potential energy within the molecules. This technique is based on the reflectance phenomenon that indicates the proportion of incident light that a given surface is able to reflect. This technique has been used for many years as a diagnostic aid in the examination of oral mucosa for the detection of OPMDs or malignant lesions [32].

The ViziLite^®^ Blue oral examination kit (Zila Pharmaceuticals, Arizona, AZ, USA) is an FDA 510(k)-cleared medical device mainly used as an adjunctive tool for visual examination of oral mucosal lesions. It is a handheld disposable chemiluminescence-based device that utilizes light illumination at wavelengths of 430 nm, 540 nm, and 580 nm inside the oral cavity. A 1% acetic acid wash is used prior to starting the light emission in order to remove surface debris and improve the visibility of epithelial cell nuclei. Abnormal oral mucosa is distinctively white (acetowhite) in appearance, whereas normal oral mucosa is lightly bluish in appearance [40,53]. A number of studies have shown varying results with the ViziLite^®^. This device is useful when detecting lesions that have not been identified by standard VOE. In particular, it is able to detect leukoplakia more accurately than erythroplakia or red lesions; this is because leukoplakia are better enhanced and visualized by this technique. However, it did not seem to detect dysplasia or cancerous lesions, regardless of whether they were red or white. Conversely, a positive ViziLite^®^ appearance does not discriminate between lesions, including keratosis, inflammation, OPMDs, ulcerated lesions, lichen planus, etc. This greatly reduces the specificity of any examination by this test and may result in many unnecessary biopsies [50,54,55,56,57].

##### Multispectral Fluorescence- and Reflectance-Based

Identafi^®^ (DentalEZ, Pennsylvania, PA, USA) is an FDA- and CE-approved, probe-like medical device designed for multispectral screening of OPMDs. Identafi^®^ uses three light sources: white light, violet light (405 nm), and green-amber light (545 nm); these can be used sequentially during an oral mucosal examination. The white light provides an optimal view for a VOE of the oral mucosa, while the violet light excites endogen fluorophores, enhancing the fluorescence of normal tissues; this results in a dark appearance when a suspicious oral lesion is present. Meanwhile, the green-amber light excites hemoglobin molecules in the blood, which allows the visualization of diffuse or dilated vasculature through reflectance spectroscopy. Thus, the third wavelength is used mainly to distinguish between benign and malignant oral lesions, with the latter being likely to have abnormal vascular structures [58]. Studies have been carried out to evaluate the oral tissue vascularity of several OPMDs using Identafi^®^ and have then compared its results with the histological grading of the biopsied lesions using a CD34 vascular marker. Detection using Identafi^®^ shows that abnormal tissues have increased vascularity compared to normal tissue, but this increase is not limited to oral cancer. This approach also detects hyperkeratosis and other lesions, such as oral lichen planus [58,59]. In another study, Identafi^®^ white light produced equivalent lesion visibility and border distinctness to that obtained using VOE. Its violet light demonstrated a specificity of 85.4%, but a low sensitivity of 12.5% when detecting oral epithelial dysplasia (OED). Finally, the green-amber light was able to identify 40.9% of lesions with vasculature visibility. Thus Identafi^®^ has potential as an aid to VOE when detecting and visualizing oral mucosal lesions via its combination of multispectral light sources. Nevertheless, it has limitations when confirming OED with good accuracy, and its clinical and optical findings should not be interpreted individually but as a whole [60].

### 2.2. New Methods under Development for Clinical Application

The role of biomarkers for oral cancer detection has emerged in recent years as a new method for clinical diagnostics. Biomarkers play an important role in distinguishing between the presence or absence of disease. They involve various types of molecules including nucleic acids, proteins, peptides, enzymatic changes, antibodies, metabolites, lipids, and carbohydrates. Biomarkers can be derived from a range specimen such as blood, serum, plasma, exfoliated cells, body secretions (sputum, saliva), or excretions (stool, urine). These specimens can be obtained through noninvasive, minimally invasive, or invasive methods [61,62].

A valid biomarker has several advantages including objective and quantitative assessment, precision of measurement, and reliability. In addition, biomarkers for cancer diseases can be used to estimate disease risk, screen for primary cancers, and distinguish between benign and malignant findings or different types of malignancy. Moreover, biomarkers can determine prognosis and predict and monitor disease status and progression as well as posttreatment disease recurrence and progression or response to therapy [61]. In contrast to the advantages of biomarkers, practical considerations and challenges of biomarkers should be considered. Measurement errors occurred in the laboratory due to improper collection, transportation, and storage of specimens. Confounding factors that may affect the measurement of biomarker should be determined beforehand. Internal factors can be age, gender, weight, and metabolic factors whereas external factors can be used in detection kit batches [63]. Cost-effectiveness is important to examine the cost and efficiency of a particular and the real impact on treatment outcome. In the past, visual screening (VOE) has been shown to be the most cost-effective approach to oral cancer screening targeting a high-risk population [64,65]. However, the evidence and systematic evaluation of biomarker cost-effectiveness in clinical practice is lacking.

At present, there are only a few clinical-validated and FDA-, CE-, or CLIA-approved biomarker tests that are available for clinical applications of oral cancer detection. These include DNA methylation biomarkers, mRNA- and protein-based expression biomarkers.

#### 2.2.1. DNA Methylation Biomarker

##### ZNF582 and PAX1

*ZNF582* belongs to a large family of transcriptional regulators that encode Krüppel-associated box zinc finger proteins (KRAB-ZFPs) [66]. A recent report has revealed that *ZNF582* acts as a tumor suppressor gene in nasopharyngeal carcinoma (NPC) by regulating the transcription and expression of the adhesion molecules *Nectin-3* and *NRXN3*. Hypermethylation of *ZNF582* promotes NPC metastasis via the regulation of these adhesion molecules [67]. The tumor suppressor role of *ZNF582* was also observed in anal cancer and esophageal carcinoma, and its methylation seems to have potential as a cancer detection biomarker [68,69]. *PAX1* is known for its paired-box domain and plays a crucial developmental role during embryogenesis [70] as well as acting as a tumor suppressor gene. A high level of *PAX1* methylation induces tumorigenesis, including oral cancer and cervical cancer [71,72,73].

Both DNA methylation biomarkers were developed as in vitro diagnostics (IVD) under the name EpiGene ZNF582 DNA Detection Kit and the EpiGene PAX1 DNA Detection Kit (iStat Biomedical, New Taipei City, Taiwan), and both are CE-approved in vitro diagnostics (IVD). This diagnostic test detects the risk level of oral precancerous lesions and oral cancers by measuring the methylation level of *ZNF582* and *PAX1*. It is an adjunctive test that assists physicians with VOE, and which helps to determine a follow-up decision as to whether to pursue a histopathological examination. This is crucial to prevent misdiagnosis when only VOE is used. The physician collects oral exfoliate cells from a suspicious lesion, and the cell specimens are then subjected to genomic DNA extraction and DNA bisulfite conversion, at which unmethylated cytosine is then converted into uracil, whereas the methylated cytosine remains unchanged. After the completion of DNA bisulfite conversion, methylation-specific PCR is then performed to detect the methylation level (methylated cytosines) of both genes.

*ZNF582* and *PAX1* were initially selected from a multi-gene panel of SCC type cancers that included *ZNF582*, *PAX1*, *SOX1*, *NKX6.1*, and *PTPRR* genes; these were later developed into a methodology using oral scrapings for the detection of oral dysplasia and oral cancer. In an earlier study by our group, oral epithelial cells were collected from 65 normal oral mucosa subjects, 107 oral precancer patients, and 95 OSCC patients. *ZNF582* methylation was able to detect mild dysplasia or oral lesions with a sensitivity and specificity of 85% and 87%, respectively. Similarly, *PAX1* methylation was able to identify moderate dysplasia or oral lesions with a sensitivity and specificity of 72% and 86%, respectively. Moreover, a high level of methylation and positivity for both genes was associated with a significant increase in disease severity. Thus, these markers may be applicable as a triage tool for patients with an abnormal result from a visual oral examination [71]. A follow-up study was implemented targeting 80 out of the 95 OSCC patients whose oral epithelial cells had been collected from the cancer and adjacent normal oral mucosal sites of the patients before cancer excision surgery. Both methylation of *ZNF582* and *PAX1* were significantly higher at the cancer sites than at the adjacent normal sites. A significantly shorter 3-year overall survival rate was found among the OSCC patients with *ZNF582* methylation positive results, *PAX1* methylation positive results, and *ZNF582*/*PAX1* methylation positive results compared with *ZNF582* methylation negative, *PAX1* methylation negative, and *ZNF582/PAX1* methylation negative patients. These findings indicate that *ZNF582* and *PAX1* methylation in cancer-adjacent normal tissue, as well as cancer sites, are potential biomarkers that can help to predict OSCC progression and OSCC patient survival [72]. Hypermethylation of *ZNF582* and *PAX1* correlates with OSCC severity among primary and recurrent OSCC patients. Both types of DNA methylation in tissues were detectable among primary and recurrent OSCC patients, and which were not influenced by age, gender, or alcohol consumption. Positive correlations were also observed between DNA-methylation level and OSCC severity, including tumor size, stage, and bone invasiveness among primary OSCC patients, as well as bone invasion among recurrent OSCC patients. In addition, methylation levels were higher among recurrent OSCC than primary OSCC patients [74].

#### 2.2.2. mRNA Biomarker

##### Multipanel mRNA OAZ1, SAT and DUSP1

OAZ1, Ornithine decarboxylase antizyme, is induced by a polyamine-dependent mechanism and has been found to be important in DNA repair and linked to the metastatic potential of human OSCC cell lines [75,76]. SAT is the rate-limiting acetyltransferase in the polyamine metabolism catabolic pathway. The enzyme catalyzes the acetylation of spermidine and spermine and is involved in regulating the intracellular concentration of polyamines and their transport out of cells. The expression of SAT has been shown to be significantly higher in prostate cancer and oral cancer [77,78]. DUSP1 is a subtype of type I cysteine-based protein tyrosine phosphatase and is involved in various signaling pathways. Salivary DUSP1 mRNA is significantly increased in OSCC patients compared with normal controls [79,80].

The three cell-free salivary mRNAs incorporated with control housekeeping genes, MT-ATP6, RPL30, were developed as a CLIA-approved test called SaliMark^TM^ OSCC test (PeriRx, Pennsylvania, PA, USA). SaliMark^TM^ OSCC test is a risk stratification test recommended to physicians when suspicious lesions have been observed. Saliva is first collected using the SaliMark^TM^ saliva collection kit; RNAs are then extracted and evaluated using reverse transcription-quantitative PCR (RT-qPCR) [81].

A case-control study of seven mRNAs (OAZ1, SAT, H3F3A, IL-1 β, IL-8, SP100P, and DUSP1) and three proteins (IL-8, M2BP, and IL-1 β) was conducted to validate whether the biomarkers are capable of discriminating patients with OSCC from healthy subjects in five independent cohorts investigated by the National Cancer Institute-Early Detection Research Network-Biomarker Reference Laboratory (NCI-EDRN-BRL). A total of 395 subjects were enrolled from five independent cohorts. All seven mRNA and three protein expression levels were increased in OSCC versus controls in all five cohorts. The expression of IL-8 and SAT were significantly increased, and both had the top sensitivity and specificity among others across the five cohorts. The validation study exhibited that these biomarkers can discriminate OSCC from a healthy control [80]. Another prospective-specimen-collection, retrospective-blinded evaluation (PRoBE) design study recommended by the National Cancer Institute (NCI) was performed by enrolling 170 patients with OSCC suspicious lesions. The aim of the study was to utilize the PRoBE study to develop new predictive mRNA models for the identification of OSCC in the intended-use patients’ population with lesions suspicious for oral cancer. The study also validated a prespecified multi-marker panel derived from prior NCI-EDRN case-control studies and validated six previously identified individual salivary mRNA markers (IL1β, IL8, OAZ1, SAT1, S100P, and DUSP1) and five housekeeping mRNAs (MT-ATP6, RPL30, RPL37A, RPL0, and RPS17) for OSCC. The RT-qPCR Ct values of individual mRNAs showed an increased concentration of approximately two-fold to nearly four-fold in OSCC. Moreover, a new model from the intended-use population incorporated with housekeeping genes demonstrated a maximal sum of sensitivity and specificity of 150.7% and an ROC curve of over 85% [77]. The validation of six prespecified individual mRNA markers of OSCC and a prespecified multi-marker model in a new prospective population supports the robustness of these markers and multi-marker methodology. New models generated in this intended-use population have the potential to further enhance the decision process for early biopsy and oral lesions at low risk, and a significantly increased risk for cancer may be identified noninvasively [77].

#### 2.2.3. Protein Biomarker

##### CD44

CD44 is a cell surface glycoprotein that is involved in many cell regulation pathways, including cell proliferation and migration [82,83]. It is also a tumor-initiating and stem cell-associated biomarker and has been shown to cause tumor initiation in different tumors, including overexpression in oral cancer [84,85]. In normal epithelium, CD44 expression occurs primarily in the basal and suprabasal region. When dysplasia progresses, CD44 expression migrates into the superficial layers, implicating its role in the early stages of carcinogenesis [86,87,88]. Furthermore, CD44 is enriched on the surface of relapsed tumors when compared to the primary tumor [89]. CD44 is cleaved and released in soluble form (solCD44) from the surface of cells by metalloproteinases that are overexpressed in advanced oral cancers [90,91].

A point-of-care (POC) IVD was developed to detect the presence of CD44 in a saliva sample to evaluate the risk of oral cancer. The BeVigilant^TM^ Rapid Test (Vigilant Biosciences, Florida, FL, USA) is a CE-approved POC IVD, which qualitatively identifies the presence of soluble CD44 (sCD44) and total protein levels in patients at risk of oral cancer. This POC is a dual-marker lateral flow test that contains two strips anchored with monoclonal antibodies for CD44 and total protein. One monoclonal captures sCD44, the other captures a different binding domain of human CD44. Incorporated with a rapid reader and the BeVigilant^TM^ Portal, physicians receive qualitative results indicating a low, moderate, or elevated level of the biomarker. A saliva sample is initially collected with 5 mL saline in a collection cup; the POC test strips are then inserted into the saliva sample until wet and are later laid flat on an absorbent pad. A visual line will appear once CD44 concentration exceeds a prespecified threshold. Concurrently, the other total protein strip changes color from yellow (no or low protein) to green (moderate range of protein) to a dark green/blue (high amount of protein) [92].

Several studies have shown the performance of detecting CD44 and total protein. In a hospital-based case-control study, 150 cases and 150 frequency-matched controls were enrolled to determine if CD44 and total protein levels in oral rinses associate with oral cancer independent of demographic and risk variables (age, gender, race, ethnicity, tobacco and alcohol use, and socioeconomic status). CD44 ≥ 5.33 ng/mL was highly associated with the case status, and total protein aided in the prediction above CD44 alone. Sensitivity and specificity in the frequency-matched control were 80% and 48.7%, respectively. However, controls were not representative of the target screening population due to 10% of these controls having a history of prior cancer. In contrast, specificity in the high-risk community was 74% and reached 95% at one-year retesting [93]. Another clinical study included 134 patients, 38 oral/oropharyngeal cancer and 96 controls. The POC was able to discriminate cancer patients from noncancerous patients with a sensitivity ranging from 71% to 84% and a specificity ranging from 30% to 50%; the NPV was 94% with a prevalence of 9.27% [92]. After an improvement was made to the POC device, a case-control cohort study was initiated on 197 patients, 67 with oral/oropharyngeal cancer and 130 controls. The detection performance exhibited 90% and 62% sensitivity and specificity, respectively. With a 34% prevalence, it showed 92% of NPV and 55% of PPV [92]. An additional laboratory ELISA test of this product, also a CE-approved IVD named OncAlert^®^ Oral Cancer LAB Test (Vigilant Biosciences, Florida, USA), provides quantitative values for CD44 and total protein. It is used as a triage test for a positive RAPID Test and uses algorithms to discriminate patients with malignancy from noncancerous normal patients. The sensitivity, specificity and NPV of the OncAlert^®^ Oral Cancer LAB Test are 81%, 93% and 98%, respectively [92].

##### S100A7

S100A7 is a calcium-binding protein and a member of the multigenic calcium-modulated S100 family; it is expressed in the upper well-differentiated spinous layer of normal epithelium [94]. Overexpression of S100A7 is found in high-risk dysplastic oral lesions linked to cancer development [95].

S100A7 is a laboratory-developed test under the name Straticyte^TM^ (Proteocyte AI, TO, Canada). This test quantitatively measures the S100A7 biomarker in biopsy tissues of patients at risk of oral cancer in conjunction with the Straticyte^TM^ proprietary algorithms, advanced imaging, and digital pathology. A biopsy specimen is first collected for histopathology examination. In parallel, the specimen can be quantitatively assessed for the level of S100A7 present in the tissue.

S100A7 was initially identified and verified through a panel of five candidate protein biomarkers, namely S100A7, prothymosin alpha (PTMA), 14-3-3ζ, 14-3-3δ, and heterogeneous nuclear ribonucleoprotein K (hnRNPK), by using proteomic approaches to distinguish oral lesions with dysplasia and oral cancers from normal oral tissues. A study was initiated to evaluate the potential of protein biomarkers candidates for identifying oral dysplastic lesions that have a high risk of cancer development. Immunohistochemistry was used to analyze the expressions of these protein biomarkers in 110 patients with biopsy-proven oral dysplasia, with known clinical outcome, and had determined their correlations with p16 expression and HPV 16/18 status. Cytoplasmic S100A7 overexpression is distinct from other protein biomarkers as the most significant candidate marker associated with cancer development in dysplastic lesions. Patients with overexpression of cytoplasmic S100A7 showed reduced oral cancer-free survival (OCFS) of 68.6 months when compared to patients with weak or no S100A7 expression in immunostaining in the cytoplasm (mean OCFS = 122.8 months) [95]. Another retrospective study was performed to develop a prognostic risk prediction model for OSCC using the five protein biomarkers. Two patient populations, Indian (test set: 282 Indian OSCCs and 209 normal tissues) and Canadian (validation set: 135 Canadian OSCC and 96 normal tissues), were recruited to analyze the correlations of expression alteration of these biomarkers with clinical and pathological parameters, and disease-free survival follow-up. A protein expression-based risk prediction model for recurrence-free survival of OSCC patient was developed based on the overall signature score. Subcellular expression of the five biomarkers was first analyzed in the Indian test set by immunohistochemistry and correlated with clinicopathological parameters and over 12 years of clinical outcome to develop a risk prediction model of recurrence-free survival. This risk prediction model was externally validated in the Canadian validation set. PTMA, S100A7 and hnRNPK had biomarker signature scores associated with recurrence-free survival of OSCC patients independent of clinical parameters. Biomarker signature scores stratified OSCC patients into a high-risk group at a median survival of 14 months and a 3-year survival rate of 30%. As for low-risk group, the survival probability did not reach 50%, and had a 3-year survival rate of 71% [96]. Straticyte^TM^, the single protein biomarker S100A7, was able to make a 5-year prediction of the probability of dysplastic lesions progressing to cancer. By comparing to histopathological dysplasia grading, Straticyte^TM^ provided greater objectivity, sensitivity, and predictive power. The sensitivity and negative predictive value (NPV) of Straticyte^TM^ were 95% and 78% (low risk vs. intermediate and high risk, respectively), whereas histopathological dysplasia was of 75% and 59% (mild vs. moderate and severe dysplasia, respectively). By quantitatively assessing S100A7, the clinical performance of Straticyte^TM^ was able to better define the risk of developing OSCC than histopathological dysplasia grading alone [97].

## 3. Developing Future Technologies

### 3.1. Artificial Intelligence (AI)-Based System

Artificial intelligence (AI) technology has grown in recent years and has been used to improve diagnoses, which can be useful for the early diagnosis of oral cancer. AI is a simulation of human intelligence and behavior via machines and now widely influences our daily lives, including medical diagnoses [98]. AI consists of two important subsets, namely machine learning (ML) and deep learning (DL). The ML algorithm is an important aspect of AI and enables computers to develop problem-solving abilities. ML typically requires an input of accurately categorized data and utilizes different types of classifiers, such as support vector machines (SVM), artificial neural networks (ANN), and decision trees, to aid computer systems during decision making [98]. This is accomplished via the development of mathematical models and accurately categorized input data and the creation of structured hierarchical learning networks [99,100]. DL relies on layers of the artificial neural networks and is modeled on behavioral patterns in the neuron layers in the neocortex of the human brain [100,101]. Thus, the depth and performance of the model depends greatly on the number of layers. DL-trained computers can process numerous algorithms efficiently and have been used to improve image interpretation, not only by shortening time and reducing the needs of specific analysis expertise, but also by extracting or correcting the essential features that can be utilized during automated medical diagnoses.

AI-based technology with clinical decision making or diagnoses systems can be useful modalities in oral cancer screening, lesion discrimination, and prediction model [102]. In the oral cancer screening application, AI has been shown to facilitate remote healthcare interactions in low- and middle-income countries. In a multistage, multicenter study, a very low-cost DL–supported smartphone-based oral cancer probe was developed for high-risk populations in rural areas with poor infrastructure [103]. The smartphone-based probe’s imaging system, combined with OSCC risk factors, provides triage guidance for the oral mucosa screener. The algorithm classified intraoral-lesion images and pairs into ‘suspicious’ and ‘not suspicious’ with sensitivity, specificity, positive predictive value, and negative predictive values ranging from 81% to 95% [104]. This study showed the potential screening effectiveness by using a smartphone incorporated with AI-based technology for healthcare providers such as general practitioners, dentists, and community workers in rural areas [105]. In another study, a tablet-based mobile microscope was developed as an oral cancer screening tool. This mobile microscope, combined with an iPad Mini with collection optics, LED illumination, and Bluetooth-controlled motors, was used to scan a slide specimen and capture high-resolution images of stained brush biopsy samples. Results demonstrated concordance between histology, cytology, and image evaluation of a remote pathologist, indicating that the device may improve screening effectiveness in rural areas and healthcare workplaces without specialists [106].

A number of studies have shown the application of AI technologies in discriminating oral lesions [107,108]. In a study by Wang et al., a partial least squares and artificial neural network (PLS-ANN) classification algorithm was utilized to discriminate the autofluorescence spectra of premalignant and malignant tissues (dysplasia and SCC) from benign tissues (healthy volunteer, OSF, and hyperkeratosis). Results showed that PLS-ANN could differentiate premalignant and malignant tissues from benign tissues with a sensitivity of 81%, specificity of 96%, and a positive predictive value of 88% [107]. In a recent study by Fu et al., they developed an automated DL algorithm using a total of 44,409 photographic images of biopsy-proven OSCC lesions and normal controls. The results demonstrated an AUC of 0.983 (95% CI 0.973–0.991), sensitivity of 94.9%, and specificity of 88.7% on the internal validation dataset. The algorithm also achieved comparable performance to that of the oral cancer specialists in terms of accuracy (92.3% vs. 92.4%), sensitivity (91.0% vs. 91.7%), and specificity (93.5% vs. 93.1%) [108].

AI technology has also been used in developing prediction models [109,110,111]. A recent study developed a personalized prediction model, web.opmd-risk.com (accessed on 15 May 2021), using an ML algorithm generated from 266 OPMDs patients in order to predict cancer risk. This model may distinguish high-risk and low-risk lesions with high sensitivity and specificity. In addition, it was able to predict the risk of future oral cancer [111]. A 15-year cohort study was conducted to validate the ML algorithm for prediction of oral cancer survival risk stratification. Extensive data from clinicopathologic and genetic data of 334 oral cancer patients was applied to ML algorithm training, which included patient characteristics, cancer site, tumor T/N stages, histopathological and surgical findings, and ultra-deep sequencing of 44 cancer related gene variant profiles. Results showed that the predictive model performed better than those using clinicopathologic data alone [109]. Traditionally, clinicopathologic markers are used by physicians to make prognostic decisions. Chang et al., investigated whether oral cancer prognosis can be improved by combining parameters of clinicopathologic and genomic markers through a hybrid of feature selection and machine learning methods. Clinicopathologic and genomic data of p53 and p63 from 31 oral cancer patients used four types of classifiers (ANN, SVM, logistic regression, and adaptive neuro-fuzzy inference system [ANFIS]). Results showed that ANFIS was the best tool for predicting oral cancer prognosis, alcohol habit, depth of invasion, and p63 were the three-input features which achieved the best accuracy (accuracy = 93.81%; AUC = 0.90). The prediction of prognosis improved using clinicopathologic and genomic data compared to clinicopathological data alone [110]. The rapid development of AI-based systems will have great impact on the screening and diagnosis of OPMDs and oral cancer. However, this technology is still in its infancy and there is a need for in-depth development prior to use as aids for diagnosis and for the prediction of oral cancer risk.

### 3.2. Lab-on-Chip

Lab-on-a-chip (LOC) is a unique microelectromechanical system that can be used to detect protein and ribonucleic acid (RNA) biomarkers. It applies microfluidics engineering technologies that integrate all aspects of a range of laboratory procedures such as specimen preparation and separation of saliva, serum, blood or urine, signal amplification, and signal detection on a single chip in an automated manner [112,113]. LOC can be used as a point-of-care device as no specialized laboratory personnel are required for the analysis and thus this approach can be easily adapted for use in remote regions or when implementing large scale screening. A wide range of molecular diagnostics, biochemical techniques, and immunoassays have been applied using LOC platforms, including nucleic acid assays, protein assays, cell sorting, etc. [112]. In a case study, 1 mL of saliva was collected and injected into a LOC device. All salivary lymphocytic cells were removed by antibodies conjugated to magnetic beads in the device, and the precancer and cancer cells were then captured and enriched using antibodies (conjugated to magnetic beads) specific for dysplasia and cancer (e.g., EpCAM) [114]. A study showed that a combination of several salivary biomarkers (proteins such as thioredoxin and IL-8 and mRNAs such as SAT, ODZ, IL-8, and IL-1b) are able to detect oral cancer with high specificity and sensitivity [115].

In general, the benefits of LOC technique during clinical diagnostics are small specimen volumes, short runtime, real-time result reporting, and automation. These advantages of LOC largely reduce reagent consumption, decrease the use of hazardous materials and exposure to infectious agents, and minimize the risk of specimen contamination. LOC also increases reproducibility and consistency, as well as generating a relatively low cost [113,114]. LOC is poised to make a significant impact on oral cancer diagnostics. However, it is currently in the early stage of development, in which there are many difficulties to address, including instant identification of various expression profiles, establishment of their predictive power, and optimization and development of appropriate specimen collection procedures.

## 4. Conclusions

Considering the size of the research and development in the field of oral cancer diagnostics, the diagnostic methods described in this article are mostly those already in clinical use or commercially available, and are showing great promises in clinical application. There are several more methods for oral cancer diagnostics that were not included in this article. Each of those diagnostic methods discloses its own uniqueness of technology in adjunctive roles of visual oral examination. Vital staining, oral cytology, and optical imaging diagnostics identify oral precancer or cancer lesions through direct optical visualization that are easy to use and create fast results. Biomarkers generate relatively objective results and possess quantitative predictive potential from the measurement of specific biological molecules. Of great promise are other potential future technologies such as AI-based systems and LOC. AI-based systems predict oral cancer via the training and analysis of large amounts of data, including images, clinicopathologic, and genetic data, whereas, LOC minimizes laboratory-scale biological molecule detection down to a small chip, both which may have a breakthrough in efficacy of oral cancer diagnosis. Greater efforts to intensify and optimize clinical performance during oral cancer diagnosis are highly encouraged to help with the effective identification of OPMDs and oral cancer.

## Figures and Tables

**Figure 1 diagnostics-11-01287-f001:**
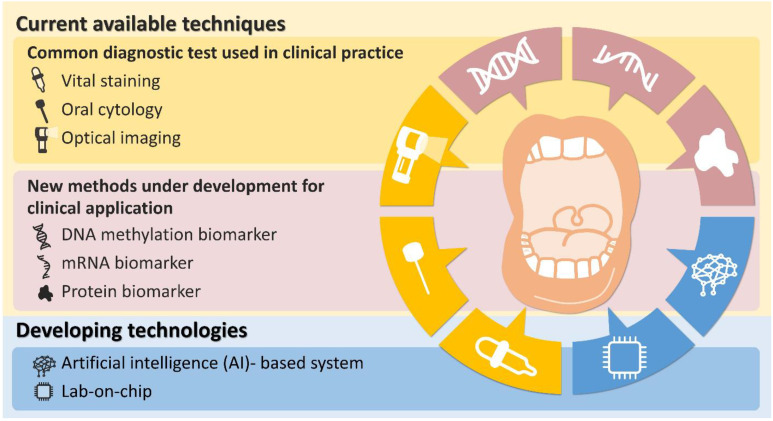
Schematic of a wide spectrum of currently available early diagnostic techniques and developing technologies for detecting oral premalignant disorders and oral cancer. The figure was created by Yi-Ju Chen for this article. Copyright © 2021 by iStat Biomedical Co., Ltd.

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
