# Peer review of "Current Insights into Oral Cancer Diagnostics"

_diagnostics, 2021, doi:10.3390/diagnostics11071287_

Round 1

Reviewer 1 Report

Dear Authors,

I have read your paper very carefully.

I have some doubts about the exposition of the methods and conclusions. I think there is a little confusion in describe currently available techniques, usefulness of ‘new’ methods in clinical practice and ‘developing technologies’ not yet subjected to validation for clinical practice (but very sponsored as such by the Authors).

Moreover, there are some important structural mistakes:

1- ‘Brush biopsy’ is not the gold standard for histological evaluation of oral biopsy. In fact, cytology cannot distinguish high grade dysplasia from carcinoma. Moreover, cytology cannot classify non-dysplastic lesions, such as lichen, leukoplakia, erythroleukoplakia, oral submucous fibrosis, proliferative verrucous, leukoplakia and erythroplakia. Surgical biopsy=histology is the gold standard for diagnosis of oral lesions and for better subclassify non-dysplastic lesions.

2- ‘Biomarker-based diagnostics’ paragraph is redundant; biomarkers listed under scientific validation. Moreover, biomarker analysis is too expensive and without evidence of cost-benefit gain in clinical practice.

3- The Authors should describe the modality of analysis, not the equipment brand. There may be a commercial conflict of interest.

4- Artificial intelligence (AI) technology is booming in recent years for improving diagnosis, particularly in early diagnosis of  oral cancer (page 8, line 282-283): please, add a reference more recent than 1995 (bibliography 101). I advice the Authors to read the paper: The contribution of artificial intelligence to reducing the diagnostic delay in oral cancer. Ilhan B, Guneri P, Wilder-Smith P. Oral Oncol. 2021 May;116:105254. doi: 10.1016/j.oraloncology.2021.105254.

5- Finally: about Bibliography:

Bibliography is not adequately formatted based on Guidelines of Diagnostics;

some citation are not accurate (n° 3: ‘WHO histological classification of tumours of Head and Neck, 4th Edition, Lyon 2017’ instead of ‘Organization, W.H. Oral Health. 2020’)

 or very inappropriate:

         87 GmbH, A.D. Prevo-Check® Rapid test for the qualitative detection of antibodies against HPV 16 L1 in whole blood and serum 2018;

  1. Prevo-Check® Rapid test for the qualitative detection of antibodies against HPV 16 L1 in whole blood and serum.

Author Response

Thank you very much for your letter of June 21, 2021, regarding our manuscript entitled” Current insights into oral cancer diagnosticsdiagnostics-1277545.  We thank you for the precious comments from reviewers, and we would like to address the concerns of the reviewers in order.

Comment/Question 1:

I have some doubts about the exposition of the methods and conclusions. I think there is a little confusion in describe currently available techniques, usefulness of ‘new’ methods in clinical practice and ‘developing technologies’ not yet subjected to validation for clinical practice (but very sponsored as such by the Authors).

Moreover, there are some important structural mistakes:

1- ‘Brush biopsy’ is not the gold standard for histological evaluation of oral biopsy. In fact, cytology cannot distinguish high grade dysplasia from carcinoma. Moreover, cytology cannot classify non-dysplastic lesions, such as lichen, leukoplakia, erythroleukoplakia, oral submucous fibrosis, proliferative verrucous, leukoplakia and erythroplakia. Surgical biopsy=histology is the gold standard for diagnosis of oral lesions and for better subclassify non-dysplastic lesions.

Authors Response 1:

(1) We thank you and appreciate very much for the valuable suggestions. We have made some structural changes and the subtopics of each section:

New structure and topic

Original structure and topic

1. Introduction

1. Introduction

2. Current available techniques 

      2.1 Common diagnostic test used in clinical              practice

            2.1.1 Vital staining

            2.1.2 Oral cytology

            2.1.3 Optical imaging

2.1.3.1 Autofluorescence-based

2.1.3.2 Chemiluminescence-based

2.1.3.3 Multispectral fluorescence-and

           reflectance-based

2.2 New methods under development for 

      clinical application

           2.2.1 DNA methylation biomarker

                     2.2.1.1 ZNF582 and PAX1

           2.2.2 mRNA biomarker

                     2.2.2.1 multi-panel mRNA OAZ1,

                                  SAT and DUSP1

          2.2.3 Protein biomarker

                    2.2.3.1 CD44

                    2.2.3.2 S100A7

2. Non-biomarker-based diagnostics

2.1 Vital staining

2.2 Brush Biopsy

2.3 Light-based techniques/Optical imaging

3. Biomarker-based diagnostics

3.1 DNA & mRNA

3.2 Protein

3. Developing future technologies

    3.1 Artificial intelligence (AI)-based system

    3.2 Lab-on-chip

4. Other potential technologies

4.1 Lab-on-chip

4.2 Artificial intelligence (AI) based system

4. Conclusions

5. Conclusions

(2) We agree and thank you for your comments that ‘Brush biopsy’ is not the gold standard for histological evaluation of oral biopsy. We have changed the topic “Brush biopsy” to “Oral cytology” and revised the content for “Oral cytology” as well in line 113-130:

“Exfoliative oral cytology is a conventional method that collects oral mucosal cells by scrapping, brushing or rinse the exfoliative cells using a tongue blade or brush [32]. The collected oral mucosal cell specimens were then fixed and stained, and their morphology was examined and interpreted by an experienced pathologist under a microscope [33]. This method is derived from a cervical Pap smear, and is simple, non-aggressive, and relatively painless [34,35]. Oral cytology was first utilized to evaluate human oral mucosal lesions in 1963. However, as a screening method for oral precancer and oral cancer, it has not achieved the same success as that of cervical cancer screening [36]. It has been shown to exhibit low sensitivity in the diagnosis of oral cancer [37,38]. This may attribute to inadequacy or nonrepresentative sampling, a high risk of procedural errors and subjectivity of interpretation by examiners [39].

Over the years, oral cytology has undergone substantial improvements for early assessment of suspicious oral lesions. Brush cytology is also a method that can be applied to individuals who have difficulties in opening their mouth for scalpel biopsies to confirm a lesion site [40]. The OralCDxÒ Brush Test (CDx Diagnostics, New York, USA) is a minimally invasive brush biopsy that is coupled with artificial intelligence computer-assisted tissue analysis. The term, brush biopsy, was however argued to be replaced by the term, oral brush cytology, as this technique should be used as a complement test and not a replace for biopsy [41]. OralCDxÒ is an advanced complementary form of exfoliative oral cytology and has an adjunctive role as a diagnostic or screening tool for the identification of OPMD at an early stage [42]. However, the clinical performance of the OralCDxÒ Brush Test remains controversial as the test's results has been found to vary significantly across several studies. Several studies reported that OralCDxÒ have high sensitivity and specificity in the detection of OSCC and precancerous lesions compared to a surgical biopsy [42-44]. However, there were also studies questioned the previously reported high performance of the OralCDxÒ system [44-46].”

Comment/Question 2:  

2- ‘Biomarker-based diagnostics’ paragraph is redundant; biomarkers listed under scientific validation. Moreover, biomarker analysis is too expensive and without evidence of cost-benefit gain in clinical practice.

Authors Response 2:

We appreciate that most of the biomarker-based diagnostic methods are either under development or optimization for manufacturing to clinical use and need to be evaluated for their cost and benefit carefully. However, current standard approaches may face challenges in some specific clinical conditions, such as large lesion area, multiple suspicious lesion loci, mass with thick verrucous or keratinized layer, and the questionable predictability to disease progression for non-dysplastic lesions. To overcome these obstacles and consider the outcome and comorbidities resulting from cancer treatment, biomarker-based diagnostics may still show its value in the early detection of risky OPMDs or early cancerous lesions. Therefore, for the integrity of this review, we keep this section, strengthen our arguments at the end of the introduction, and also make some context changes as following. (1) We have changed the topic of the “biomarker-based diagnostics” paragraph to “New methods under development for clinical application”. Also, the subtopics were changed accordingly (refer to Authors Response 1).

(2) More clinical study information of each biomarker was described in each biomarker section:

2.2.2.1 multi-panel mRNA OAZ1, SAT and DUSP1 in line 266-284

“A case-control study of 7 mRNA (OAZ1, SAT, H3F3A, IL-1 β, IL-8, SP100P and DUSP1) and 3 proteins (IL-8, M2BP and IL-1 β) was conducted to validate whether the biomarkers are capable of discriminating patients with OSCC from healthy subjects in 5 independent cohorts investigated by National Cancer Institute-Early Detection Research Network-Biomarker Reference Laboratory (NCI-EDRN-BRL). A total of 395 subjects were enrolled from 5 independent cohorts. All 7 mRNA and 3 protein expression level were increased in OSCC versus controls in all five cohorts. The expression of IL-8 and SAT were significantly increased and both had the top sensitivity and specificity among others across the 5 cohorts. The validation study exhibited that these biomarkers are able to discriminate OSCC from healthy control [80]. Another prospective-specimen-collection, retrospective-blinded evaluation (PRoBE) design study recommended by National Cancer Institute (NCI) was performed by enrolling 170 patients with OSCC suspicious lesions. The aim of the study was to utilize the PRoBE study to develop new predictive mRNA models for the identification of OSCC in the intended-use patients’ population with lesions suspicious for oral cancer. The study also validated a pre-specified multi-marker panel derived from prior NCI-EDRN case-control studies and validated 6 previously identified individual salivary mRNA markers (IL1β, IL8, OAZ1, SAT1, S100P, and DUSP1) and 5 housekeeping mRNAs (MT-ATP6, RPL30, RPL37A, RPL0, and RPS17) for OSCC. The RT-qPCR Ct values of individual mRNAs showed an increased concentration of approximately 2-fold to nearly 4-fold in OSCC. Moreover, a new model from the intended-use population incorporated with housekeeping genes has demonstrated a maximal sum of sensitivity and specificity of 150.7% the ROC curve of over 85% [77]. The validation of 6 pre-specified individual mRNA markers of OSCC and a pre-specified multi-marker model in a new prospective population supports the robustness of these markers and the multi-marker methodology. New models generated in this intended-use population have the potential to further enhance the decision process for early biopsy and oral lesions at very low risk and at significantly increased risk for cancer could be identified noninvasively [77].”

2.2.3.1 CD44 in line 305-320

“Several studies have shown the performance of detecting CD44 and total protein. In a hospital-based case-control study, 150 cases and 150 frequency-matched controls were enrolled to determine if CD44 and total protein levels in oral rinses associates with oral cancer independent of demographic and risk variables (age, gender, race, ethnicity, tobacco and alcohol use, and socioeconomic status). CD44 ≥5.33 ng/ml was highly associated with the case status, and total protein aided prediction above CD44 alone. Sensitivity and specificity in the frequency-matched control was 80% and 48.7%, respectively. However, controls were not representative of the target screening population due to 10% of these controls had history of prior cancer. In contrast, specificity in the high-risk community was 74% and reached 95% at one-year retesting [93]. Another clinical study included 134 patients, 38 oral/oropharyngeal cancer and 96 controls. The POC was able to discriminate cancer patients from non-cancerous patients with a sensitivity ranging from 71% to 84% and a specificity ranging from 30% to 50%; the NPV was 94% with a prevalence of 9.27% [92]. After an improvement was made to the POC device, a case-control cohort study was initiated on 197 patients 67 with oral/oropharyngeal cancer and 130 controls. The detection performance exhibited a 90% and 62% of sensitivity and specificity. With 34% of prevalence, it showed 92% of NPV and 55% of PPV [92]. An additional laboratory ELISA test of this product, also a CE-approved IVD, named OncAlert® Oral Cancer LAB Test (Vigilant Biosciences, Florida, USA), provides quantitative values for CD44 and total protein. It is used as a triage test for a positive RAPID Test, and uses algorithms to discriminate patients with malignancy from non-cancerous normal patients. The sensitivity, specificity and NPV of the OncAlert® Oral Cancer LAB Test are 81%, 93% and 98%, respectively [92].“

2.2.3.2 S100A7 in line 329-354

“S100A7 was initially identified and verified through a panel of 5 candidate protein biomarkers namely S100A7, prothymosin alpha (PTMA), 14-3-3ζ, 14-3-3δ and heterogeneous nuclear ribonucleoprotein K (hnRNPK) by using proteomics approaches to distinguish oral lesions with dysplasia and oral cancers from normal oral tissues. A study was initiated to evaluate the potential of those protein biomarkers candidate for identify oral dysplastic lesions at high risk of cancer development. Immunohistochemistry was used to analyze the expressions of those protein biomarkers in 110 patients with biopsy-proven oral dysplasia, with known clinical outcome, and had determined their correlations with p16 expression and HPV 16/18 status. Cytoplasmic S100A7 overexpression standouts from other protein biomarkers as the most significant candidate marker associated with cancer development in dysplastic lesions. Patients with overexpression of cytoplasmic S100A7 showed reduced oral cancer-free survival (OCFS) of 68.6 months when compared to patients with weak or no S100A7 expression in immunostaining in the cytoplasm (mean OCFS = 122.8 months)[95]. Another retrospective study was performed to develop a prognostic risk prediction model for OSCC using the 5 protein biomarkers. Two patient populations, Indian (test set: 282 Indian OSCCs and 209 normal tissues) and Canadian (validation set: 135 Canadian OSCC and 96 normal tissues), were recruited to analyze the correlations of expression alteration of these biomarkers with clinical and pathological parameters, and disease-free survival follow-up. A protein expression-based risk prediction model for recurrence-free survival of OSCC patient was developed based on the overall signature score. Subcellular expression of the 5 biomarkers was first analyzed in the Indian test set by immunohistochemistry correlated with clinicopathological parameters and over 12 years of clinical outcome to develop a risk prediction model of recurrence-free survival. This risk prediction model was externally validated in the Canadian validation set. PTMA, S100A7 and hnRNPK had biomarker signature score associated with recurrence-free survival of OSCC patients independent of clinical parameters. Biomarker signature score stratified OSCC patients into high-risk group at a median survival of 14 month, and had 3-year survival rate of 30%. As for low-risk group, the survival probability did not reach 50%, and had 3-year survival rate of 71% [96]. StraticyteTM, the single protein biomarker S100A7, was able to make a 5-year prediction of the probability of dysplastic lesions progressing to cancer. By comparing to histopathological dysplasia grading, StraticyteTM provided greater objectivity, sensitivity, and predictive power. The sensitivity and negative predictive value (NPV) of StraticyteTM were 95% and 78% (low risk vs. intermediate and high risk), whereas, histopathological dysplasia was of 75% and 59% (mild vs. moderate and severe dysplasia). By quantitatively assessing S100A7, the clinical performance of StraticyteTM was able to better define the risk of developing OSCC than histopathological dysplasia grading alone [97].”

(3) The advantage and disadvantages of biomarkers, and the cost-effectiveness was described in line 197-208:

“A valid biomarker has several advantages including objective and quantitative assessment, precision of measurement and reliable. In addition, biomarkers for cancer diseases can be used to estimate disease risk, for screening primary cancers, for distinguishing benign from malignant findings or one type of malignancy from another. Moreover, biomarkers can determine prognosis, predicts and monitor disease status and progression as well as post-treatment disease recurrence and progression or response to therapy [61]. In contrast to advantages of biomarkers, some practical considerations and challenges of biomarkers should be considered. Measurement error occurred in the laboratory due to improper collection, transportation and storage of specimens. Confounding factors that may affect the measurement of the biomarker should be determined beforehand. Internal factors can be age, gender, weight, metabolic factors whereas external factors can be the used of detection kit batches[63]. Cost-effectiveness is important to examine the cost and efficiency of a particular and the real impact on the treatment outcome. In the past, the visual screening (VOE) has been shown to be the most cost-effective approach to oral cancer screening targeting to the high-risk population [64,65]. However, the evidence and systematic evaluation of biomarker cost-effectiveness in clinical practice is lacking.”

Comment/Question 3:  

3- The Authors should describe the modality of analysis, not the equipment brand. There may be a commercial conflict of interest.

Authors Response 3:

(1) We agree with reviewer’s comments and have played down the description of test products in brand names to avoid the suspicion of commercial conflict of interest. We then supplemented in more detail of the principle, clinical intended use, regulatory status and clinical evidence available of each technology. Our revision is as below:

2.1.1 Vital staining in line 98-105

“Toluidine blue has been shown to have high sensitivity, but relatively low specificity [27,28]. A recent hospital-based diagnostic accuracy study was carried out to evaluate the efficacy of Toluidine blue staining which was served as an adjunct tool to standard clinical examination to facilitate early detection of oral cavity and oropharynx malignant lesions. 55 subjects with OPMDs or malignant lesions were subjected to detailed clinical examination and toluidine blue staining. By comparing the staining results with histopathological examination, Toluidine blue test detected malignancy with a sensitivity of 92.6% and a specificity of 67.9%, respectively. The overall diagnostic accuracy was 80%. The result indicates that toluidine blue staining method can be a valuable adjunctive diagnostic process for oral and oropharyngeal cancers[27]. In practice, it is a useful way of identifying lesions with possible malignant changes.“

2.1.2 Oral cytology in line 113-130

“Exfoliative oral cytology is a conventional method that collects oral mucosal cells by scrapping, brushing or rinse the exfoliative cells using a tongue blade or brush [32]. The collected oral mucosal cell specimens were then fixed and stained, and their morphology was examined and interpreted by an experienced pathologist under a microscope [33]. This method is derived from a cervical Pap smear, and is simple, non-aggressive, and relatively painless [34,35]. Oral cytology was first utilized to evaluate human oral mucosal lesions in 1963. However, as a screening method for oral precancer and oral cancer, it has not achieved the same success as that of cervical cancer screening [36]. It has been shown to exhibit low sensitivity in the diagnosis of oral cancer [37,38]. This may attribute to inadequacy or nonrepresentative sampling, a high risk of procedural errors and subjectivity of interpretation by examiners [39].

Over the years, oral cytology has undergone substantial improvements for early assessment of suspicious oral lesions. Brush cytology is also a method that can be applied to individuals who have difficulties in opening their mouth for scalpel biopsies to confirm a lesion site [40]. The OralCDxÒ Brush Test (CDx Diagnostics, New York, USA) is a minimally invasive brush biopsy that is coupled with artificial intelligence computer-assisted tissue analysis. The term, brush biopsy, was however argued to be replaced by the term, oral brush cytology, as this technique should be used as a complement test and not a replace for biopsy [41]. OralCDxÒ is an advanced complementary form of exfoliative oral cytology and has an adjunctive role as a diagnostic or screening tool for the identification of OPMD at an early stage [42]. However, the clinical performance of the OralCDxÒ Brush Test remains controversial as the test's results has been found to vary significantly across several studies. Several studies reported that OralCDxÒ have high sensitivity and specificity in the detection of OSCC and precancerous lesions compared to a surgical biopsy [42-44]. However, there were also studies questioned the previously reported high performance of the OralCDxÒ system [44-46].”

2.1.3.1 Autofluorescence-based in line 138-152

“Autofluorescence is refer to a light of particular wavelength interacts with cells, and causes excitation and re-emission of light of different wavelengths. Autofluorescence emitted from tissues is produced by fluorophores that naturally occurs in the human tissues, for instance collagen, tryptophan, elastin, keratin, hemoglobin and NADH etc., are the naturally occurring fluorophores. The concentration of these fluorophores alters in OPMDs and cancerous lesions which cause alterations of natural light scattering and absorption properties of the tissues [48].

VELScopeÒVx (LED Dental, British Columbia, Canada) is a CE-approved medical device that non-invasively screens for alterations in oral mucosal autofluorescence [49]. It is a handheld camera device that emits blue light (400 nm and 460 nm wavelengths) that is combined with proprietary optical filtering in order to visualize oral abnormalities. The normal oral mucosa produces green autofluorescence light, while anomalous oral mucosal lesions absorb the autofluorescent light and appear as dark areas that contrast with the adjacent tissue [49],[50]. One clinical study has shown that use of the VELScopeÒVx is beneficial for confirming the presence of OPMDs including erythroplakia, leukoplakia, and other oral tissue disorders; however, it is unable to discriminate between low-risk and high-risk lesions [51]. Another study found that the VELScopeÒ Vx alone performed relatively poorly when detecting epithelial dysplasia, albeit while elevated oral mucosal lesions were visible and during the process a few clinically undetected lesions were uncovered. This suggests the VELScopeÒVx is more suitably used when accompanied by relevant clinical interpretation or by other methods for the diagnosis of epithelial dysplasia [52].”

2.1.3.2 Chemiluminescence-based in 155-170

“Chemiluminescence is referred to the emission of visible range of light radiation after the electrons, excited by a chemical exergonic reaction, returning from the excited to the ground state; light photons are released upon the transitions of electronic potential energy within the molecules. This technique is based on the reflectance phenomenon that indicates the proportion of incident light that a given surface is able to reflect. This technique has been used for many years as a diagnostic aid in the examination of oral mucosa for the detection of OPMDs or malignant lesions[32].

The ViziLiteÒ Blue oral examination kit (Zila Pharmaceuticals, Arizona, USA) is a FDA 510(k)-cleared medical device mainly used as an adjunctive tool for the visual examination of oral mucosal lesions. It is a handheld disposable chemiluminescence-based device that utilizes light illumination at wavelengths of 430 nm, 540 nm and 580 nm inside the oral cavity. A 1% acetic acid wash is used prior to the starting the light emission in order to remove surface debris and improves the visibility of epithelial cell nuclei. Abnormal oral mucosa is distinctively white (acetowhite) in appearance, whereas normal oral mucosa is lightly bluish in appearance [40,53]. A number of studies have shown varying results with the ViziLiteÒ. This device is useful when detecting lesions that have not been identified by standard VOE. In particular, it is able to detect leukoplakia more accurately than erythroplakia or red lesions; this is because leukoplakia are better enhanced and visualized by this technique. However, it did not seem to be able to detect dysplasia or cancerous lesions, regardless of whether they are red or white. On the other hand, a positive ViziLiteÒ appearance does not discriminate lesions including keratosis, inflammation, OPMDs, ulcerated lesions, lichen planus and so on. This greatly reduces the specificity of any examination by this test and may result in many unnecessary biopsies [50,54-57].“

Comment/Question 4:  

4- Artificial intelligence (AI) technology is booming in recent years for improving diagnosis, particularly in early diagnosis of oral cancer (page 8, line 282-283): please, add a reference more recent than 1995 (bibliography 101). I advise the Authors to read the paper: The contribution of artificial intelligence to reducing the diagnostic delay in oral cancer. Ilhan B, Guneri P, Wilder-Smith P. Oral Oncol. 2021 May; 116:105254. doi: 10.1016/j.oraloncology.2021.105254.

Authors Response 4:

Thank you very much for recommending the useful literature. We have read the paper and enriched the content in section “3.1 Artificial intelligence (AI)- based system” in line 371-409:

“AI based technology with clinical decision-making or diagnosis systems can be useful modalities in oral cancer screening, lesion discrimination and prediction model [102]. In oral cancer screening application, AI has been shown to facilitate remote healthcare interactions in low- and middle-income countries. In a multistage, multicenter study, a very low-cost DL–supported smartphone-based oral cancer probe was developed for high-risk populations in rural areas with poor infrastructure [103]. The smartphone-based probe’s imaging system combined with OSCC risk factors and provides triage guidance for the oral mucosa screener. The algorithm classified intraoral lesions images and pairs into ‘suspicious’ and ‘not suspicious’ with sensitivity, specificity, positive predictive value, and negative predictive value ranging from 81% to 95% [104]. This study showed the potential screening effectiveness by using smartphone incorporated with AI-based technology for healthcare providers such as general practitioner, dentist or even community worker in rural areas [105]. In another study, a tablet-based mobile microscope was developed as an oral cancer screening tool. This mobile microscope combined with an iPad Mini with collection optics, LED illumination, and Bluetooth-controlled motors to scan a slide specimen and capture high-resolution images of stained brush biopsy samples. Results demonstrated concordance between histology, cytology, and image evaluation of remote pathologist, indicating that the device may improve screening effectiveness in rural areas and healthcare workplace without specialists [106].

A number of studies have shown the application of AI technologies in discriminating oral lesions [107,108]. In a study by Wang et. al., a partial least squares and artificial neural network (PLS-ANN) classification algorithm was utilized to discriminate the autofluorescence spectra of premalignant and malignant tissues (dysplasia and SCC) from benign tissues (healthy volunteer, OSF and hyperkeratosis). Results showed that PLS-ANN could differentiate premalignant and malignant tissues from benign tissues with a sensitivity of 81%, specificity of 96%, and a positive predictive value of 88% [107]. In a recent study by Fu et. al., have developed an automated DL algorithm using a total of 44,409 photographic images of biopsy-proven OSCC lesions and normal controls. The result demonstrated an AUC of 0.983 (95% CI 0.973–0.991), sensitivity of 94.9%, and specificity of 88.7% on the internal validation dataset. The algorithm also achieved comparable performance to that of the oral cancer specialists in terms of accuracy (92·3% vs 92.4%), sensitivity (91·0% vs 91·7%), and specificity (93·5% vs 93·1%) [108].

AI technology has also been used in developing prediction model [109-111]. A recent study has developed a personalized prediction model, web.opmd-risk.com, using a ML algorithm generated from 266 OPMDs patients in order to predict cancer risk. This model could distinguish high-risk and low-risk lesions with high sensitivity and specificity. In addition, it also was able to predict the risk of future oral cancer [111]. A 15-year cohort study was conducted to validate the ML algorithm for prediction of oral cancer survival risk stratification. Extensive data from clinicopathologic and genetic data of 334 oral cancer patients was applied for the ML algorithm training, which includes patient characteristics, cancer site, tumor T/N stages, histopathological and surgical findings, and ultra-deep sequencing of 44 cancer related gene variant profiles. Results showed that the predictive model performed better than those using clinicopathologic data alone [109]. Traditionally, clinicopathologic markers are used by physicians to make prognostic decisions. Chang et. al., investigated whether oral cancer prognosis can be improved by combining parameters of clinicopathologic and genomic markers through a hybrid of feature selection and machine learning methods. Clinicopathologic and genomic data of p53 and p63 from 31 oral cancer patients using 4 types of classifiers (ANN, SVM, logistic regression, and adaptive neuro-fuzzy inference system [ANFIS]). Results showed that ANFIS was the best tool for predicting oral cancer prognosis, and alcohol habit, depth of invasion, and p63 were the 3-input features which achieved the best accuracy (accuracy = 93.81%; AUC = 0.90). The prediction of prognosis improved using clinicopathologic and genomic data compared to clinicopathological data alone [110]. The rapid development of AI-based systems will have great impact on the screening and diagnosis of OPMDs and oral cancer. However, this technology is still in its infancy and there is a need for in-depth development prior to use as aids to diagnosis and for the prediction of oral cancer risk.”

Comment/Question 5:  

5- Finally: about Bibliography:

Bibliography is not adequately formatted based on Guidelines of Diagnostics; some citation is not accurate (n° 3: ‘WHO histological classification of tumors of Head and Neck, 4th Edition, Lyon 2017’ instead of ‘Organization, W.H. Oral Health. 2020’)

 or very inappropriate:

 87 GmbH, A.D. Prevo-Check® Rapid test for the qualitative detection of antibodies against HPV 16 L1 in whole blood and serum 2018;

  1. Prevo-Check® Rapid test for the qualitative detection of antibodies against HPV 16 L1 in whole blood and serum.

Authors Response 5:

(1) Thank you for the comments. We have revised the bibliography “Organization, W.H. Oral Health. 2020” to “WHO histological classification of tumors of Head and Neck, 4th Edition, Lyon 2017” in line 463.

(2) The content of a protein biomarker (Prevo-CheckÒ) of the following bibliography was removed, therefore they were excluded from the reference section:

“87 GmbH, A.D. Prevo-Check® Rapid test for the qualitative detection of antibodies against HPV 16 L1 in whole blood and serum 2018” and

“88. Prevo-Check® Rapid test for the qualitative detection of antibodies against HPV 16 L1 in whole blood and serum.”

In summary, we appreciate all the thoughtful suggestions provided from the reviewers.

Please find a revision of the review paper which includes a clean version and a track-changed version. The changes that we have made in the manuscript, at the request of the reviewers, have strengthened the manuscript and resulted in a more meaningful and enlightening manuscript.  Many of our responses in this letter are cited in the revised text, in appropriate portions of the revised Abstract, Introduction, Content and Conclusion sections as well as the Figure. 

We are confident that the content of this revised manuscript with rigorous and

important message has been delivered clearly. We sincerely hope that the revised version would be suitable for publication in Diagnostics.   

Truly yours,

Yee-Fun Su

Yi-Ju Chen

Fa-Tzu Tsai

Wan-Chun Li

Ming-Lun Hsu

Ding-Han Wang

Cheng-Chieh Yang

Reviewer 2 Report

This is a review on the subject of oral cancer diagnostics.

MAJOR COMMENTS

In the Abstract it is said the following about tissue biopsies: ...due to several limitation and inaccuracy of this method...". That is a severe statement. What is meant by that? What data support this statement?

The language needs severe corrections.

Author Response

Thank you very much for your letter of June 21, 2021, regarding our manuscript entitled” Current insights into oral cancer diagnosticsdiagnostics-1277545.  We thank you for the precious comments from reviewers, and we would like to address the concerns of the reviewers in order.

Comment/Question 1:

In the Abstract it is said the following about tissue biopsies: ...due to several limitation and inaccuracy of this method...". That is a severe statement. What is meant by that? What data support this statement?The language needs severe corrections.

Authors Response 1:

(1) Thank you for your comment. The original purpose of this article is to have an overview of the current available techniques for the clinical diagnosis of OPMDs and oral cancer in an adjunctive manner. To avoid misleading, the gold-standard role of tissue biopsy followed by histopathological examination to get a definite diagnosis has been emphasized in the first paragraph of introduction. However, current standard approaches may face challenges in some specific clinical conditions, such as large lesion area, multiple suspicious lesion loci, mass with thick verrucous or keratinized layer, and the questionable predictability to disease progression for non-dysplasia lesions. The conventional visual oral examination (VOE) as the first step for oral cancer screening heavily depends on the experience and proficiency of the clinicians. These introduced solutions may assist the clinicians to get objective evaluation of suspicious lesions, provide appropriate clinical care, and improve the treatment outcome. This argument has been strengthened in the first paragraph of the introduction and other contexts. The wording of the sentence and description in the earlier version was somewhat inappropriate and misleading, so we have revised the sentence as following in line 21-24:

Before revision:

““To date, a visual oral examination remains the routine first line method of identifying oral lesions, and tissue biopsy remains the gold standard for OSCC diagnosis. However, this approach has a number of limitations and can be inaccurate; as a result, some patients are diagnosed when their cancer has reached a severe stage or a high-risk patient with OPMD is misdiagnosed and left untreated.”

After revision:

“To date, a visual oral examination remains the routine first line method of identifying oral lesions, however, this method lies certain limitations, as a result, some patients are diagnosed when their cancer has reached a severe stage or a high-risk patient with OPMD is misdiagnosed and left untreated.”

We have forwarded our manuscript for English language revision as suggested by reviewer.

(2) The limitations of VOE are described in line 51-56:

“At present, a visual oral examination (VOE) is the routine screening method used to identify oral mucosal lesions. However, VOE depends heavily on the experience of the physician, owing the fact that some OPMDs such as white or red lesions, persistent ulcers are often indistinguishable at its clinical presentation [17]. A summary of global oral cancer forum in 2016 depicted the shortcomings of VOE, that the result of VOE is dependent on the quality of examiner. It requires training and calibration of screeners, is indistinguishable between benign lesions, cancer and OPMDs and etc.[18].”

The limitations of biopsy histopathological examination are described in line 64-74:

“Despite the well-established and systemic approaches of the histopathologic examination, some clinical conditions may face the clinician’s great challenges to get an accurate diagnosis. First, in a carcinogen-exposed oral cavity, the suspicious lesions commonly span across a large area or scatter at multiple loci. The mucosa affected by the effects of field cancerization or several lesions under various cancer-developing status may require multiple biopsies. Nevertheless, due to its invasiveness, many patients fear taking repeated biopsies, regardless of the presence of symptomatic or asymptomatic lesions. In addition, for some well-differentiated cancers or verrucous leukoplakias, the thickness and heavy keratinization of the lesions usually hinders the biopsy procedures to get a deep enough representative tissues specimen. Therefore, the histopathological diagnosis heavily relies on the location, size, depth and quality of the biopsied specimen, the techniques of fixation and freezing used on the biopsy, and the physician’s professional background and experience. The clinical observer’s subjectivity and variability during the VOE and histopathological examination will pose limitations on OPMD detection and oral cancer diagnosis [24,25].”

In summary, we appreciate all the thoughtful suggestions provided from the reviewers.

Please find a revision of the review paper which includes a clean version and a track-changed version. The changes that we have made in the manuscript, at the request of the reviewers, have strengthened the manuscript and resulted in a more meaningful and enlightening manuscript.  Many of our responses in this letter are cited in the revised text, in appropriate portions of the revised Abstract, Introduction, Content and Conclusion sections as well as the Figure. 

We are confident that the content of this revised manuscript with rigorous and

important message has been delivered clearly. We sincerely hope that the revised version would be suitable for publication in Diagnostics.   

Truly yours,

Yee-Fun Su

Yi-Ju Chen

Fa-Tzu Tsai

Wan-Chun Li

Ming-Lun Hsu

Ding-Han Wang

Cheng-Chieh Yang

Round 2

Reviewer 1 Report

Well done. The paper has been improved.

Reviewer 2 Report

The authors have met and responded to previous comments in a satisfactorty way.